# Comparative Analysis and Identification of Terpene Synthase Genes in *Convallaria keiskei* Leaf, Flower and Root Using RNA-Sequencing Profiling

**DOI:** 10.3390/plants12152797

**Published:** 2023-07-28

**Authors:** Sivagami-Jean Claude, Gurusamy Raman, Seon-Joo Park

**Affiliations:** Department of Life Sciences, Yeungnam University, Gyeongsan 38541, Gyeongbuk, Republic of Korea; sivajeankor@gmail.com

**Keywords:** transcriptome, differential gene expression, floral volatile, monoterpenes, sesquiterpenes

## Abstract

The ‘Lilly of the Valley’ species, *Convallaria*, is renowned for its fragrant white flowers and distinctive fresh and green floral scent, attributed to a rich composition of volatile organic compounds (VOCs). However, the molecular mechanisms underlying the biosynthesis of this floral scent remain poorly understood due to a lack of transcriptomic data. In this study, we conducted the first comparative transcriptome analysis of *C*. *keiskei*, encompassing the leaf, flower, and root tissues. Our aim was to investigate the terpene synthase (TPS) genes and differential gene expression (DEG) patterns associated with essential oil biosynthesis. Through de novo assembly, we generated a substantial number of unigenes, with the highest count in the root (146,550), followed by the flower (116,434) and the leaf (72,044). Among the identified unigenes, we focused on fifteen putative *ckTPS* genes, which are involved in the synthesis of mono- and sesquiterpenes, the key aromatic compounds responsible for the essential oil biosynthesis in *C*. *keiskei*. The expression of these genes was validated using quantitative PCR analysis. Both DEG and qPCR analyses revealed the presence of *ckTPS* genes in the flower transcriptome, responsible for the synthesis of various compounds such as geraniol, germacrene, kaurene, linalool, nerolidol, trans-ocimene and valencene. The leaf transcriptome exhibited genes related to the biosynthesis of kaurene and trans-ocimene. In the root, the identified unigenes were associated with synthesizing kaurene, trans-ocimene and valencene. Both analyses indicated that the genes involved in mono- and sesquiterpene biosynthesis are more highly expressed in the flower compared to the leaf and root. This comprehensive study provides valuable resources for future investigations aiming to unravel the essential oil-biosynthesis-related genes in the *Convallaria* genus.

## 1. Introduction

Plant secondary metabolites (PSMs) are essential defense mechanisms that benefit human health [1]. Secondary metabolites, including essential groups such as phenolics, terpenes and nitrogen-containing compounds, are frequently lineage-specific and enable plants to interact with the biotic and abiotic environment [2]. Plant terpenoids are the most abundant secondary metabolites, exhibiting varied structures and aromatic scents [3]. The most common chemical structure of terpenes is characterized by the number of five-carbon isoprene units in the skeleton structure [4,5]. This includes isoprene (C_5_), monoterpenes (C_10_), sesquiterpenes (C_15_), diterpenoids (C_20_), sesterterpenes (C_25_), triterpenes (C_30_) and tetraterpenes (C_40_) [6]. Additionally, terpenoids mainly play a critical role in plant defense, pollinator attraction and interactions between plants and their environment, either directly or indirectly [7]. Specifically, floral scents consist of monoterpenes such as linalool, limonene, myrcene and trans-β-ocimene, which are responsible for the biosynthesis of essential oils and determine the quality of the scents, as well as sesquiterpenes including farnesene, nerolidol and caryophyllene components [8]. Terpenoid biosynthesis mainly depends on two independent pathways: the mevalonic acid (MVA) pathway and the methylerythritol phosphate (MEP) pathway [9]. The MVA pathway mainly involves the biosynthesis of sterols, sesquiterpenes and triterpenes, while the MEP pathway involves mono- and diterpene biosynthesis in plants [10]. The ample catalytic versatility of terpene synthase (TPSs) in plants accounts for the vast variety of volatile terpenoids. The terpene synthases are able to synthesize multiple products from prenyl diphosphate precursors, including geranyl pyrophosphate (GPP), farnesyl diphosphate (FPP) and geranylgeranyl diphosphate (GGPP) [11]. Various *TPS* genes associated with floral scent biosynthesis have been progressively reported in *Citrus sinensis* [12], *Daucus carota* [13], *Camellia sinensis* [14] and *Lathyrus odoratus* [15]. The TPS family genes represent medium to large-sized families, with almost all plant species containing approximately 20–150 TPS-like functional genes [16]. Based on a systematics analysis, TPS is divided into seven subfamilies: TPS-a, b, c, d, e/f, g and h classes [16]. TPS-b encodes terpene synthase with an R(R)X_8_W motif that acts as an isomerization catalyst in angiosperms. TPS-c and TPS-g are ancestral synthases in gymnosperm that synthesize mono-, di-, and sesquiterpenes. Kaurene synthase is required to produce gibberellic acid under the TPS-e/f class [17]. Additionally, TPS-g encodes for monoterpenes lacking the R(R)X8W motif; this type of TPS produces acyclic monoterpenes contributing to floral VOCs. Several studies have used genome-wide approaches to uncover TPS in rice, *Arabidopsis*, grapes, carrots, apples, eucalyptus, tea tree and tomatoes [14,18,19,20,21,22,23,24].

The Asparagaceae family belongs to the flowering plants within the Asparagales order of the monocots [25]. *Convallaria* is a small genus of monocotyledons in the subfamily Nolinoideae that comprises three species: *C. keiskei*, *C. majalis*, and *C. montana* [26]. Since there are no significant morphological differences among these three species, they have been classified based on their geographical distributions [26,27]. *C. keiskei* is also called ‘the Asian Lilly of the Valley’ and is only distributed in Korea, Japan and China [28]. *Convallaria* possesses potent cardioactive glycosides and functions as a geophyte, rendering it toxic to herbivores [29]. These geophytic toxins represent a promising reservoir of bioactive metabolites with substantial potential for both health-related and agronomic applications [30]. In addition, *Convallaria* is a bell-shaped flower with a sweet aroma and is extensively used in fragrances and other cosmetic products owing to the presence of volatile compounds [31,32]. Notably, as early as 1765, a French florist employed *Convallaria* as a perfumed floral product, suggesting the existence of multiple volatile organic compounds (VOCs) in all parts of the plant [33]. 

Hwang et al. [34] analyzed the presence of volatile constituents present in the *C*. *keiskei* flower using a GC-MS approach. They identified four terpene alcohols, accounting for 47.91% of the volatile composition, followed by ten saturated hydrocarbons (24.41%), three monoterpenes (5.8%), thirteen unsaturated hydrocarbons (9.16%), and an additional twenty-two compounds, including acids, alcohols, aldehydes, esters, ketones and terpene esters, comprising 12.4% of the total composition. Among these compounds, the primary volatile terpene constituents found in *C*. *keiskei* flowers were dihydromyrcenol (17.85%), *trans*-geraniol (17.14%), nerol (12.78%) and pentadecane (11.91%). These compounds are known to contribute significantly to the floral scents associated with *C*. *keiskei*. In contrast, thus far, no molecular studies have been conducted to comprehensively investigate the genes associated with volatile biosynthesis in *C*. *keiskei*. More recently, several genomic studies have focused on unraveling the genome structure and evolutionary aspects of *Convallaria* [26,27]. Notably, *Convallaria* is the first genus to provide evidence of mitochondrial DNA transfer to the chloroplast genome in the Asparagales order, making it an excellent model plant for studying genome evolution [26,27]. However, despite these advances, to the best of our knowledge, no transcriptomic studies have been conducted to elucidate the genes involved in essential oil synthesis in *C. keiskei*. Additionally, recent transcriptome analysis has been widely used to identify the genes involved in floral scent metabolites in some plant species [35,36,37,38,39]. Therefore, in the present study, we conducted a comparative analysis of the leaf, flower and root of *C. keiskei* using transcriptomic analyses to understand the genes involved in essential oil biosynthesis. In addition, we performed an analysis of *TPS* genes in *C*. *keiskei* to discover and gain knowledge of their contribution in a tissue-specific manner. Our findings reveal distinct patterns of *TPS* genes in the leaf, flower and root, with tissue-specific expression patterns. This indicates that one *TPS* gene may be involved in synthesizing multiple mono- and sesquiterpenes in *C. keiskei*. This study is the first comprehensive transcriptomic analysis of *C. keiskei*, providing transcriptomic data on which may further enrich the understanding of the molecular regulation and biosynthesis of terpenoid-related genes, which are the primary component of *C. keiskei* essential oil in the genus.

## 2. Materials and Methods

### 2.1. RNA Isolation, cDNA Library Construction, and Illumina Sequencing

The total RNA from the leaf, flower and root parts of *C*. *keiskei* was extracted using the protocol outlined by Bretial et al. [40]. To perform RNA-sequencing, cDNA libraries were constructed, and the sequencing was carried out using the Illumina HiSeq 2500 platform at Phyzen Ltd., South Korea. Each sample yielded ~15–16 GB of raw data, consisting of 44.65 million, 41.33 million and 42.86 million paired-end (2 × 150 bp) reads for the flower, leaf, and root of *C. keiskei*, respectively. The average insert size of the generated reads was ~550 bp.

### 2.2. De Novo Assembly and Functional Annotation

The quality assessment of the raw reads was analyzed using FastQC v0.11 [41], and low-quality reads were removed using Trimmomatic v0.39 [42]. The resulting clean reads were then assembled to retrieve unique transcripts using the trinity v2.13.2 with a *Kmer* size of 25, employing the inchworm, chrysalis and butterfly pipelines [43]. A schematic diagram of the methodology is provided in Appendix A.

Subsequently, the unique transcripts were subjected to coding function annotations using the ‘Trinotate and TrinotateWeb’ pipeline, as previously described [44]. The gene annotations for all the assembled unigenes was aligned against the Swiss-Prot database (https://www.uniprot.org/), accessed on 4 February 2022 and the Kyoto Encyclopedia of Genes and Genomes (KEGG, http://www.genome.jp/kegg/kegg2.html) database accessed on 4 February 2022. A complete protein family (Pfam) [45] analysis was performed using HMMER [46].

Gene ontology terms were analyzed using the Trinotate pipeline (https://github.com/Trinotate/Trinotate.github.io/wiki, accessed on 4 February 2022) and TRAPID tools (http://bioinformatics.psb.ugent.be/trapid_02/) accessed on 4 February 2022 to retrieve the biological process (BP) terms. Furthermore, the completeness of the transcripts was assessed using BUSCO, utilizing tblastn alignments against the eudicots database lineage (eudicots_odb10) [47].

### 2.3. Identification and Phylogenetic Tree Analysis of Terpene Synthase Unigenes

The peptide sequence of the *A*. *thaliana* database protein was downloaded from the TAIR website [48]. The terpene synthase unigenes were identified by conducting a local blastX search against the NCBI-Blast tool [49] and the KEGG database [50], with a threshold value of 1 × 10^5^. Pathway prediction was carried out using the BioCyc database (https://biocyc.org/ARA/organism-summary?object=ARA) accessed on 10 February 2022.

To identify floral VOC-specific *TPS*, *TPS* genes from *Mentha, Vitis*, *Nicotiana* and Ocimum were used [51,52]. Additionally, *Convallaria* unigenes were employed as a reference database for blasting against the total transcripts to predict the TPSs for further verification. The conserved domain of TPS was identified using MEME [53].

The structural variance of *TPS* was assessed by aligning the protein sequences using MAFFT with default parameters, followed by constructing a maximum likelihood tree using PHYML (LG) with 100 bootstrap replications [54,55].

### 2.4. Differentially Expressed Gene (DEG) Analysis

The transcripts obtained from the assembly of whole sequences of *C*. *keiskei* were analyzed for differentially expressed genes (DEGs) using RNA-Seq by Expectation-Maximization (RSEM). Initially, an index was generated, and then the total transcripts were processed and aligned to the raw reads of *C*. *keiskei* flower, leaf and root samples [56].

The RSEM method was used to calculate the fragments per kilobase per million (FPKM) and transcripts per million (TPM) values for the expressed or abundant unigenes specific to *C*. *keiskei* tissues. The DEGs were identified based on FDR-corrected options with a *p*-value <  0.01 and log_2_ fold change (log_2_FC) > 1. The FPKM and TPM values were utilized to estimate the expression levels of the unigenes involved in the terpenoid pathway and volatile compound biosynthesis.

The complete DEG analysis was performed using the Trinity pipeline [43], incorporating Bowtie2 [56,57] and EdgeR [58]. The significance of the DEGs was determined based on thresholds of FDR  ≤  0.01 and |log_2_FC|  ≥  1. Visualization of the results was carried out using ggplot2 (https://ggplot2.tidyverse.org/reference/geom_point.html, 4 February 2022).

### 2.5. Quantative PCR Analysis in C. keiskei

The expression levels of MVA-PEP biosynthesis-related unigenes in the *C. keiskei* leaf, flower and root samples were quantified using the FPKM values. Gene-specific primers were designed based on the FPKM data and used for quantitative PCR (qPCR) analysis (Appendix A). KEGG annotation and BLAST analysis of the transcriptome leaf data were utilized to identify the unigenes involved in the terpenoid backbone pathway. The complete open reading frame (ORF) of the selected unigenes was determined to design forward and reverse primers for further analysis. In addition to the TPS genes, seven candidate unigenes and random MEP/MVA unigenes were chosen based on amplicon size and primer structure (Appendix A). For cDNA synthesis, a reverse transcription system (A3500, Promega (product made in USA), Yuseong-gu, Daejeon, Korea) was used, and 20 µL of the final cDNA product was incubated at 70 °C for 5 min with random primers. The quantification assay was performed using a GoTaq^®^ qPCR master mix (A6001, Promega) under standard cycling conditions, with triplicate samples used as a guideline (Applied Biosystems 7500 step one plus). The significance of the gene expression was calculated using a t-test in R (https://www.rdocumentation.org/packages/stats/versions/3.6.2/topics/t.test, accessed on 30 April 2022).

## 3. Results

### 3.1. De Novo Assembly and Annotation Stats for C. keiskei

RNA-sequencing libraries were constructed from the leaf, flower, and root to perform transcriptome analysis and study the distribution of terpenoid synthase genes and their by-products in different tissues of *C. keiskei*. The raw reads obtained from these libraries were deposited in the NCBI-SRA database under accession numbers SRR20982553-55. The highest number of unigenes (146,550) was obtained from the root, followed by the flower (116,434), and the lowest number of unigenes was observed in the leaf (72,044). The quality of the assembled transcripts was assessed using BUSCO, which revealed that the flower had 99% complete transcripts, while the leaf and root had 98% complete transcripts. The duplicated gene content was also identified. The overall assessment and annotation analysis indicated that the transcripts were of good quality (Appendix A). The *A. thaliana* database from TAIR was used for analysis and annotation, and approximately 20% of the genes from the transcriptome database were retained. Annotation of the protein family database (Pfam) annotated approximately 30% to 52.2% of the genes. KEGG analysis was performed to identify pathways associated with the unigenes, and 40% to 51.03% of the unigenes were successfully assigned to pathways. The contigs from the three transcriptomes were integrated and de novo assembled, resulting in a total of 237,129 unigenes. These assembled unigenes were then annotated using BLAST against three public databases (TAIR, KEGG and Pfam). Out of the total unigenes, 138,029 (35.17%) were successfully annotated, with at least one functional gene in the leaf, flower and root of *C. keiskei* (Table 1).

### 3.2. Gene Ontology and KEGG Annotation

In the leaf tissue of *C*. *keiskei*, a gene ontology (GO) category study revealed 182 biological process terms. Among these, 51.73% of the unigenes were involved in the organic substance metabolic process, while 45.64% were categorized under the primary metabolic pathway. In terms of the secondary metabolic processes, 2.23% of the genes were associated with chlorophyll metabolism, 1.4% with carotenoid metabolism and 0.88% with xanthophyll metabolism, and 3.32% were found to respond to jasmonic acid.

For the flower tissue, 110 GO categories were identified. Of these, 22.28% of the genes were involved in the developmental process, 21.62% in anatomical structural development and 12.08% in reproduction. In terms of secondary metabolic categories, 1.25% of the genes were associated with flavonoid metabolism and 0.65% with iso-pentenyl diphosphate (IPP) metabolism.

In the root tissue, 198 biological function categories were observed. Among these, 28.34% of the unigenes were involved in chemical responses, while 20.38% were involved in inorganic substance responses. The root unigenes also exhibited secondary metabolic processes related to salicylic and jasmonic acid synthesis, with 1.08% of the genes involved in ethylene metabolism.

When considering the combined assembled unigenes from all tissues, 228 GO categories were shared. Among these, 59.13% were related to metabolic process. In the plant secondary metabolic process, 3.18% were involved in isoprenoid metabolism, and 2.94% were associated with terpenoid metabolism (Appendix A).

Through the KEGG annotation, it was found that terpene cyclases from linear oligoprenyl diphosphate precursors were responsible for the production of the largest class of natural products. The complete assembly of *C*. *keiskei* generated 237,129 unigenes, of which 36.9% (68,282 unigenes) were identified in different routes through KEGG analysis (Appendix A). Overall, 1047 of the unigenes were associated with metabolic pathways, and 165 of the unigenes were related to the biosynthesis of secondary metabolisms in diverse environments.

The terpenoid backbone biosynthesis analysis revealed 31 databases matched to the KEGG database in the combined *C*. *keiskei* unigenes. The pathway involved the production of various terpenoids, starting from acetyl-CoA to IPP, which produced FPP and GPP, leading to the production of different terpenoids. In the monoterpenoid pathway, compounds such as linalool and neomenthol were found in the flowers and leaves of *C*. *keiskei* (Figure 1). In the diterpenoid pathway, eleven types of Gibberellic acid (GA_53,44,19,20,29,51,9,24,15,12,34_) were identified in the flower, leaf and root of *C. keiskei*. Sesquiterpenoid oils, including nerolidol, squalane and germacrene B, were found in the flower samples, while nerolidol, neomenthol, germacrene B and the triterpenoid oil amyrin were present in the leaf samples. The root transcriptome exhibited a higher abundance of monoterpenoids such as limonene, citronellal, citronella, oxo geranial, caryophyllene and germacrene B compared to the leaf and flower (Figure 1).

### 3.3. Transcriptome-Wide Identification of TPS Genes Using Phylogenetic Tree Analysis

In the search for putative *TPS* genes in *C. keiskei*, *Arabidopsis TPS* genes were initially used as a query. Subsequently, a *C. keiskei TPS* gene was identified and used as an additional query to find more *C*. *keiskei TPS* genes. As a result, twenty-nine genes were further identified. These genes were analyzed for complete open reading frames (ORF) and conserved domains, with an average length of ~566 amino acids (Appendix A). The systematic analysis of the identified *TPS* genes revealed that they can be categorized into five classes. Among these, four unigenes were in the TPS-a class, three in TPS-b, two in TPS-c, one in TPS-e/f and ten in TPS-g classes. Additionally, six sesquiterpene-involved TPS genes were identified (Figure 2). The TPS-c, TPS-e and TPS-f classes are known to be involved in the biosynthesis of di-terpenes, mono-terpenes, and some sesquiterpenes. Out of the different TPS classes identified, only 15 unigenes had complete ORFs compared to the other retained unigenes, and they exhibited conserved motifs (Appendix A).

### 3.4. Identification of Floral Scent-Related ckTPS

The floral scent or volatile organic compound (VOC)-related TPS in *C. keiskei* were identified using a phylogenetic analysis with known functional genes from other species’ databases. The complete open reading frames (ORFs) of 15 *TPS* genes were analyzed, and their association with known VOC compounds was determined by comparing them with functional genes from the *Arabidopsis*, *Menth*, *Vitis*, *Nicotiana* and *Ocimum* species (Appendix A). As a result, we found four *ckTPS*, including the *ckTPS6*, *14*, *17* and *18* genes, associated with the *S*-linalool synthase. A *ckTPS2* was associated with geraniol, and two genes, *ckTPS12* and *29,* were involved in farnesol production. *ckTPS28* was associated with neomenthol, a compound present in the leaf and flower of *C. keiskei*. Other *ckTPS1*, *3*, *9* and *20*, were related to the putative triterpenoid synthase members, and *ckTPS11* was encoded with trans-ocimene, which was highly expressed in the leaves and roots. Further analysis using the Swiss protein database revealed the expression of *ckTPS23* (germacrene synthase), *ckTPS4*, *5* and *10* (trans-ocimene synthase), *ckTPS13* (nerolidol synthase), *ckTPS16*, *27*, *28* (Ent-kaur-16-ene synthase) and *ckTPS6* and *18* (*S*-linalool synthase) in the flower samples (Appendix A). Additionally, four putative *TPS* genes with unknown TPS members were identified, representing the presence of novel and unknown functional metabolites in *C. keiskei* (Appendix A).

### 3.5. Differential Expression Analysis of the Leaf, Flower and Root Samples of C. keiskei

An analysis of differentially expressed unigenes in the leaf, flower and root of *C. keiskei* using the Trinity pipeline revealed a total of 3152 genes that showed significant differential expression (Appendix A). Among these genes, 956 unigenes were significantly upregulated in the leaf, 236 in the flower and 417 in the root, and 50 unigenes were significantly upregulated in all the tissues (Figure 3A). Additionally, 367 unigenes were upregulated in both the flower and leaf, while 187 unigenes were upregulated in both the flower and root of *C. keiskei*. On the other hand, 471 genes showed down-regulation from the leaf to the flower (Figure 3B). The analysis of differentially expressed genes between the root and flower identified 1001 significantly downregulated genes, with 259 unigenes showing downregulation from the root to leaf. An annotation analysis of the top differentially expressed genes revealed that the upregulated unigenes in the leaf were mainly related to the photosynthesis process and phenylpropanoid biosynthesis in *C. keiskei*.

In the flowers, the upregulated unigenes were associated with lipid biosynthesis, terpene synthase genes, and the auxin response. In the root, the upregulated unigenes were primarily involved in organic chemical compound metabolism and terpene synthase genes (Figure 4; Appendix A). The downregulation analysis showed that the genes involved in lipid biosynthesis were downregulated in the leaf. Genes related to methyltransferase, metallothionein and glycohydrolase 32 classes in the flower were highly downregulated. In the root, genes involved in glucosidase, GAP-dehydrogenase, photosynthesis and chalcone–flavanone isomerase were downregulated (Figure 4; Appendix A). The volcano plots of the differentially expressed *TPS* genes revealed tissue-specific expression patterns in *C. keiskei,* highlighting the novel functions of *ckTPS2*, *ckTPS7*, *ckTPS14* and *ckTPS23* (Figure 4). These four *TPS* genes showed significant expression differences in the comparative analysis and exhibited conserved motifs (Figure 4; Appendix A). Among them, *ckTPS2* and *ckTPS23* were upregulated in the flower and downregulated in the leaf and root. *ckTPS14* was significantly upregulated in the leaf and downregulated in the flower and root. *ckTPS7* showed high upregulation in the root and downregulation in the leaf and flower of *C*. *keiskei*. *ckTPS2*, belonging to the TPS-g class, is potentially related to geraniol synthase involved in the production of mono-terpenoids responsible for the floral scent in *C. keiskei* flowers. *ckTPS23*, belonging to the TPS-a class, is upregulated in flowers, and may play a role in unknown floral scent functions in *C*. *keiskei*.

### 3.6. Terpene Backbone Pathway and Terpene Synthase Expression Patterns in C. keiskei

A comparative transcriptome profiling of the *C. keiskei* terpene backbone pathway revealed the presence of essential enzymes involved in the MVA and MEP pathways. In the MVA pathway, the *ACAT* key component, which has five paralogous copies, was upregulated in a tissue-specific manner (Figure 5). The *HMG synthase1* (*HMGS1*) gene showed upregulation in the leaf and flower, while *HMGS2* was upregulated in the root. One copy of *HMGR* was upregulated in the flower but downregulated in the leaf and root of *C*. *keiskei*. (Figure 5). In the MEP pathway, the *DXS* gene was highly upregulated in the flower but downregulated in the leaf and root. Two copies of the *DXR* unigene were upregulated in the flower and leaf but downregulated in the root (Figure 5). Four copies of *HDS* and one copy of *HDR* were upregulated in both the leaf and flower but downregulated in the root tissue. The *ISFD* or *IDI* gene showed upregulation in the leaf and root but not in the flower of *C. keiskei*. The *TPS* by-production genes *GGPP* and *FFPP*, which lead to the production of various types of *TPS*, were found in one and two copies, respectively. The *FFPP1* and *FFPP2* in the MVA pathway were upregulated in the leaf and flower but downregulated in the root tissue. In contrast, the *GGPP* unigene was upregulated in the flower and downregulated in the root and leaf of *C. keiskei* (Figure 5).

A comparative analysis of *TPS* expression patterns among the leaf, flower and root of *C*. *keiskei* showed that the flower and root had a higher distribution of *TPS* genes, while fewer *ckTPS* genes were found in the leaf (Figure 6A). The expression pattern among the samples did not produce a DE value; therefore, the expected count versus *FPKM* of *TPS* unigenes from the RSEM analysis was examined. The flower exhibited higher expression of the top eight *ckTPS* genes, with significant FPKM values. The leaf showed the highest FPKM values for six *ckTPS* genes, and the root showed significant expression values for two *ckTPS* (Figure 6A).

A Venn diagram plot showed the common sharing of TPS genes among *C*. *keiskei*, with 17 *ckTPS* genes distributed among 28 unique *ckTPS* (Figure 6B). Six *ckTPS* genes were shared between the flower and root, and two *ckTPS* genes were unique to the flower and leaf. One unique *ckTPS* gene was found in the leaf and root (Figure 6B). However, most of the flower *ckTPS* genes showed a more significant expression compared to the flower and root. While the root had the highest number of *ckTPS* genes, their expected count was lower than that in the flower (Figure 6). Overall, this study revealed unique and significantly expressed *ckTPS* genes in the flower compared to the leaf and root of *C*. *keiskei*. The expression patterns of 15 *TPS* unigenes, which had complete ORFs, were examined and compared among the leaf, flower, and root of *C*. *keiskei* (Appendix A). When comparing the leaf and flower, most of the *ckTPS* genes were downregulated, and only four *TPS* genes were upregulated in the *C*. *keiskei* leaf. Four unique *TPS* genes were upregulated when comparing the leaf and root. The majority (nine *ckTPS* genes) of the upregulated genes were observed in the flower and root of *C*. *keiskei* (Appendix A).

### 3.7. Quantitative Assay of Terpene Backbone Pathway Genes in C. keiskei

Quantification studies revealed that the expression of the key gene, *ACAT* was significantly higher in the flower of *C. keiskei* compared to the leaf and root. *ACAT* plays a crucial role in the terpene backbone pathway by converting acetyl CoA into acetyl aceto CoA in the cytoplasm. Similarly, the expression pattern of HMGS, the regulatory gene in the terpene backbone pathway, was notably significant in the flower and root of *C*. *keiskei*, where it converts acetyl aceto CoA into mevalonate (Figure 7).

Within the MEP pathway, the *DXS* gene is located in the plastids and encodes 1-deoxy-D-xylulose 5-phosphate synthase (DXS), a key enzyme essential for the pathway. *DXS* showed significant expression in the leaf, flower, and root of *C*. *keiskei*, indicating its importance in terpene biosynthesis across different tissues. Furthermore, qPCR was performed to validate the identified *TPS* genes, including the top four *TPS* genes from the DEG analysis.

The qPCR results confirmed that *ckTPS11* exhibited significant expression in all the examined tissues of *C. keiskei* (Figure 7). *ckTPS14* demonstrated significant expression in the flowers compared to the leaf and root. *ckTPS16* showed significant expression in the leaf, with slightly lower levels in the flower and root. *ckTPS2* displayed unique expression in all the tissues of *C*. *keiskei*. *ckTPS23* exhibited expression in the flower and root, while *ckTPS6* showed significant expression in the root and leaf, indicating a tissue-specific expression pattern in *C*. *keiskei*. Furthermore, *ckTPS7* showed higher expression in flowers compared to the leaf and root of *C. keiskei* (Figure 7).

## 4. Discussion

*Convallaria*, a genus belonging to the Nolinoideae subfamily, is renowned for its captivating floral scent and rich assortment of medicinal compounds. It encompasses only three known species worldwide [27,28]. Terpene synthase (TPS) enzymes play a pivotal role in the production of significant essential oils in plants [11], while also governing sugar metabolism, embryonic development and response to abiotic stress [1,59]. The intricate floral scent, comprised of hundreds of volatile organic compounds (VOCs), serves the dual purpose of attracting pollinators and deterring herbivores [60]. In recent years, there has been a surge of interest in exploring floral fragrance, specifically in unraveling the biosynthesis, emission and regulatory mechanisms of floral volatiles [61]. TPS enzymes serve as the primary catalysts for the production of monoterpenes (C_10_), sesquiterpenes (C_15_) or diterpenes (C_20_) from their respective substrates: geranyl diphosphate (GPP), farnesyl diphosphate (FPP) or geranylgeranyl diphosphate (GGPP) [11]. However, there is a dearth of reports or studies investigating the molecular mechanisms underlying essential oil biosynthesis-related genes in the genus *Convallaria*. Notably, earlier studies have indicated that monoterpenes, the key components responsible for essential oil biosynthesis, dominate various parts of the *C*. *keiskei* plant [34]. Hence, the primary objective of this study was to identify tissue-specific volatile compound-related and *TPS*-related genes by leveraging a well-established TPS protein database and employing a differential gene expression analysis in *C. keiskei*. In this study, we employed a transcriptomic approach to sequence leaf, flower, and root tissues of *C. keiskei*, utilizing the de novo assembly method to retain tissue-specific unigenes and facilitate gene expression analysis, thereby unraveling essential oil biosynthesis-related genes. Additionally, we conducted a transcriptome-wide identification of *TPS* genes in *C*. *keiskei*, shedding light on VOC-related genes and other putative *TPS*-related genes. Notably, plant VOC terpenes play a critical in floral scents and confer protection against environmental stresses [62], encompassing fatty and amino acids that contribute to floral and fruit aromas [63]. The elucidation of the key genes involved in volatile biosynthetic pathways provides valuable insights into the molecular mechanisms governing floral fragrance regulation.

The mevalonate pathway has gained significant commercial value due to its ability to produce desirable by-products, such as monoterpenes, which find extensive use in the cosmetics and pharmaceutical industries [64]. In this regard, the researchers employed specific precursors, including mevalonate and 1-deoxy-D-xylulose, to evaluate the relative involvement of this pathway [65]. It has been established that the plastid–methylerythritol phosphate (MEP) pathway primarily generates monoterpenes, such as linalool, geraniol and other components [66]. Conversely, the mevalonate (MVA) pathway in plants produces sesquiterpenes such as nerolidol, farnesol and other related metabolites [67]. The current study focused on elucidating the expression pattern of genes related to both the MVA and MEP pathways in *C. keiskei,* indicating the production of mono- and sesquiterpenes as volatile organic compounds (Figure 1). Notably, the TPS enzyme, which utilizes FPP as its primary substrate, is encoded by two gene copies, suggesting an alternative pathway for sesquiterpene production in *Convallaria* that occurs in both the cytosol and plastid compartments. Additionally, regulatory enzymes associated with GGPP synthesis via the plastid–MEP pathway contribute to the synthesis of monoterpenes (Figure 1). Earlier studies have also indicated that monoterpenes are significant contributors to the production of floral volatile organic compounds in plants [68]. Consistent with these findings, the present study observed higher gene expression levels of genes involved in monoterpenes synthesis in *C*. *keiskei*.

Using transcriptomic approaches, a total of 28 unique *ckTPS* genes were successfully identified in *C*. *keiskei*. These *ckTPS* genes were classified based on their amino acid length and the presence of different subfamily motif types. Through a conserved domain analysis and phylogenetic tree analysis, the *ckTPS* genes were categorized into five TPS subfamilies (TPS-a, b, c, g and e/f) (Figure 2). Among these *ckTPS* genes, 15 were found to be functional and responsible for the synthesis of various mono-, sesqui-, and other terpenoid compounds in *C*. *keiskei*. Notably, *ckTPS9* and *ckTPS12* in the TPS-a and TPS-b clades lacked the R(R)X_8_W motif, while one member of the *ckTPS16*-e/f subfamily also lacked this motif (Appendix A). Furthermore, the sesquiterpene synthase TPS family included *ckTPS8*, which did not possess the DXXXD domain. These findings align with previous studies by Chen et al. [16] and Liu and Fu [69], where *TPS*-related family genes lacking specific motifs or domains were identified using transcriptomic approaches. It is worth mentioning that genome-wide identification of *TPS* genes has been reported in several land plants, including *Oryza sativa*, *A. thaliana*, *Vitis vinifera*, *Daucus carota*, *Malus domestica*, *Eucalyptus*, *Camellia sinensis* and *Solanum lycopersicum* [14,18,19,20,21,22,23,24]. In the present study, we successfully identified *TPS*-related genes in *C*. *keiskei,* and future investigations will deepen our understanding of the molecular mechanisms and relationships between essential oil synthesis-related genes within the genus *Convallaria* and its closely related species.

Furthermore, we identified the expression of three *ckTPS* (*ckTPS2*, *7* and *14*) and one sesquiterpene-related (*ckTPS23*) gene in all three *C*. *keiskei* tissues using DEG analysis. Additionally, we performed qPCR analysis to validate our results. The qPCR analysis confirmed the high expression levels of three other *ckTPS*-related genes (*ckTPS6*, *11* and *16*) in all three parts of *C*. *keiskei*. These genes exhibited the presence of R(R)X8W and DXXXD domains, which were also identified in sweet basil and lavender plants [70,71]. Among these genes, *ckTPS2* and *ckTPS23* showed significant upregulation in the flower tissue compared to the leaf and root tissues (Figure 3). Further blast annotation of *ckTPS2* revealed its similarity to the geraniol synthase (GES) gene, which is responsible for the sedative effects of floral odor [33]. Expression analysis showed upregulation of *ckTPS2* in the flowers (log_2_FC = 4.806) and downregulation in the leaf (log_2_FC = −2.188) and root (log_2_FC = −2.618) tissues of *C*. *keiskei* (Figure 4). A quantification analysis confirmed the significant expression of *ckTPS2* in all the tissues of *C. keiskei* (Figure 7*)*, indicating its involvement in the synthesis of geraniol, an acyclic monoterpene that contributes to floral fragrance with its light, refreshing and floral-woody-citrus notes [72]. Geraniol is abundant in flowers with distinctive structures, and finds applications in fragrances, pharmaceuticals and agrochemical industries [73].

The DEG analysis also revealed distinct log_2_FC values for *ckTPS23* in the flower, leaf and root tissues, with values of +2.27, −1.27 and −2.61, respectively. Quantification studies further confirmed that *ckTPS23* exhibited significant expression in the flowers of *C*. *keiskei*, while showing comparatively lower expression levels in the leaf and root tissues (Figure 6 and Figure 7). *ckTPS23* shares an 88% amino acid sequence identity with the *V. vinifera*-Germacrene D (Ger D) synthase gene [74]. Hence, our findings suggest the synthesis of *Ger D* throughout the plant, as supported by the KEGG pathway analysis. Ger D synthase, a tricyclic sesquiterpene using FPP as a substrate, has been reported in various developmental stages, and decreases during flower opening in ylang-ylang flowers [75].

Similarly, rose transcriptome and headspace analyses identified *Ger D* as a major emission contributing to floral scent [76]. In *Camellia*, *TPS* gene analysis revealed paralogous copies of *Ger D* synthase associated with cold assimilation [14]. Therefore, our study indicates the production of geraniol and *Ger D* compounds as volatile components of *Convallaria* flowers. Additionally, *ckTPS4*, *5* and *10* were found to be involved in the synthesis of trans-ocimene: acyclic monoterpenes that give citrus, green and woody aromas [77]. These genes showed expression in both leaf and flower tissues and play roles in attracting floral visitors and defending against herbivory [78].

*ckTPS13*, identified as nerolidol synthase, utilizes FPP and GPP as substrates to produce acyclic nerolidol and linalool through a bifunctional enzyme in plants [79]. *ckTPS13* exhibited unique and significant expressions in the flowers, while its distribution in the leaf and root tissues was limited. In *Lilium*, linalool is the primary monoterpene responsible for floral scents [77]. On the other hand, (E)-Nerolidol is a sesquiterpene that is abundant in the essential oils of various plants, including *Piper claussenianum*, *Momordica charantia*, *Ginglo biloba*, *Baccharis dracunculifolia*, *Zanthoxylum hyemale*, *Zorniabra siliensis* and *Swinglea glutinosa*, as well as many flowering plants such as lavender, neroli, lemongrass, tea tree and ginger [80,81,82]. It is known for its aroma and possesses antioxidant, pharmacological and biological properties [83]. (E)-Nerolidol is considered an effective insecticide against *Stevia* (Aphididae) and tea trees (*Pediculus*) [84,85].

In the leaf tissue, *ckTPS14* showed significant upregulation in DE analysis and downregulation in the flower and root tissues of *C*. *keiskei*. However, quantification studies have revealed its significant expression in the flower and root, and relatively minor expression in the leaf (Figure 6 and Figure 7). *ckTPS14* has been identified as an *S*-linalool synthase in *Magnolia grandiflora* through genomic characterization studies [86]. Both DEGs and quantification analysis have indicated linalool synthesis in all tissue types. GPP serves as a substrate for *ckTPS14*, producing monocyclic alcohols, including linalool, in *Vitis* and *Pinus* spp. [87]. Along with *ckTPS14* (leaf), *ckTPS6*, 17 and *18*, which represent identical copies of *S*-linalool synthase, showed relatively low expression in flowers but contributed to the synthesis of various VOC compounds. Thus, *ckTPS14* in the leaf and *ckTPS6*, *17* and *18* in the flower are associated with α-terpineol biosynthesis in *C*. *keiskei*. Furthermore, *ckTPS16*, *ckTPS27* and *ckTPS28* were identified as Ent-kaur-16-ene synthase A (or Kaurene synthase [KS]), involved in the synthesis of Ent-kaur-16-ene, a diterpenoid and a key enzyme in gibberellic acid biosynthesis found in plants, fungi and bacteria [88]. These genes exhibited moderate expression in all *Convallaria* tissues, indicating the synthesis of KS. Ent-kaur-16-ene is formed by the catalytic action of ent-KS and ent-copalyl diphosphate (ent-CPP), using GGPP as a substrate through bifunctional diterpene synthase in *Cucurbita maxima* [89,90].

In the root tissue of *C*. *keiskei*, *ckTPS7*, representing valencene synthase (VS), exhibited significant expression. VS is a class 1 plant terpene cyclase or synthase that converts FPP into sesquiterpene valencene, which is widely used in perfumes and cosmetics. However, the quantification analysis of *ckTPS7* showed significant expression in the flower and negligible expression in the root and leaf tissues of *C. keiskei* (Figure 6 and Figure 7). The quantification results differed from the DEGs studies. VS, a sesquiterpene compound, has been identified as a significant volatile compound emitted in flowers of *Vitis* and *Citrus* [74]. A study on *Celastrus* indicated higher VS compound synthesis in roots based on a transcriptome analysis [91]. Therefore, our findings suggest that VS is involved in the floral odor of *C*. *keiskei* flowers and has a role in root development. Additionally, downregulation of trans-ocimene synthase (*ckTPS11*) biosynthesis was observed in the *C*. *keiskei* roots. A quantification analysis of *ckTPS11* showed significant root and leaf expression but low expression in flowers. The DEG analysis showed upregulation only in root tissues, indicating trans-ocimene synthesis in *C*. *keiskei*.

Previous research has demonstrated that certain monoterpene and sesquiterpene compounds, including *S*-linalool, α-terpineol, citronella, citronellate, 8-oxo geranial, limonene, β-caryophyllene and α-humulene, play essential roles in floral scent production, plant defense against herbivory and antioxidative activities [33]. In the present study, highly expressed monoterpenes and sesquiterpenes were identified in the monocot plant *C*. *keiskei*, particularly in the flowers, compared to the leaves and roots (Appendix A). These terpenes are primarily involved in floral scent production. The identification of *TPS* genes through genome-wide and transcriptome sequencing provides valuable insights into the unique genes responsible for essential oil or floral VOC synthesis. This information contributes to the genomic resources available for future studies on non-model plants. The transcriptomic data obtained for *C. keiskei* will expedite research in functional genomics and the differential expression of genes across various tissues, such as leaves, flowers and roots. The identified unigenes and their transcriptional profiles, particularly those involved in TPS synthesis pathways, enhance our understanding of the molecular mechanisms underlying TPS biosynthesis. Furthermore, this knowledge will facilitate future efforts to engineer terpene biosynthesis in the *Convallaria* genus.

## 5. Conclusions

The genus *Convallaria* lacks functional gene resources for the commercially significant non-model plant. In this pioneering study, we sequenced the leaf, flower and root tissues of *C*. *keiskei* and performed de novo transcriptome assembly using high-quality RNA sequencing data. Comparative analyses were conducted to identify the transcripts associated with terpene biosynthesis and VOC-related genes in ‘Lilly of the Valley’ flowers, leaves and roots, and their expression patterns were confirmed through quantitative PCR analysis. The analysis of differentially expressed (DE) genes revealed that most *TPS* unigenes, which encode terpene synthases, exhibited higher expression in the flowers compared to the leaves and roots. Notably, the terpene synthase-related genes found in *C*. *keiskei* included mono- and sesquiterpene synthases, consistent with the essential oils previously reported using the headspace technique. This study is a crucial steppingstone for discovering the *ckTPS* gene family and provides valuable insights for future genetic investigations in *C. keiskei*. Moreover, it lays the foundation for metabolic engineering endeavors targeting VOC-related genes in *C*. *keiskei* flowers. In summary, our study fills the knowledge gap regarding gene resources in *Convallaria* and significantly contributes to understanding terpene biosynthesis in *C*. *keiskei*. It facilitates the discovery of *ckTPS*-related gene families and provides essential information for future genetic studies. Furthermore, it opens up new possibilities for metabolic engineering strategies focusing on VOC-related genes in ‘Lilly of the Valley’ flowers.

## Figures and Tables

**Figure 1 plants-12-02797-f001:**
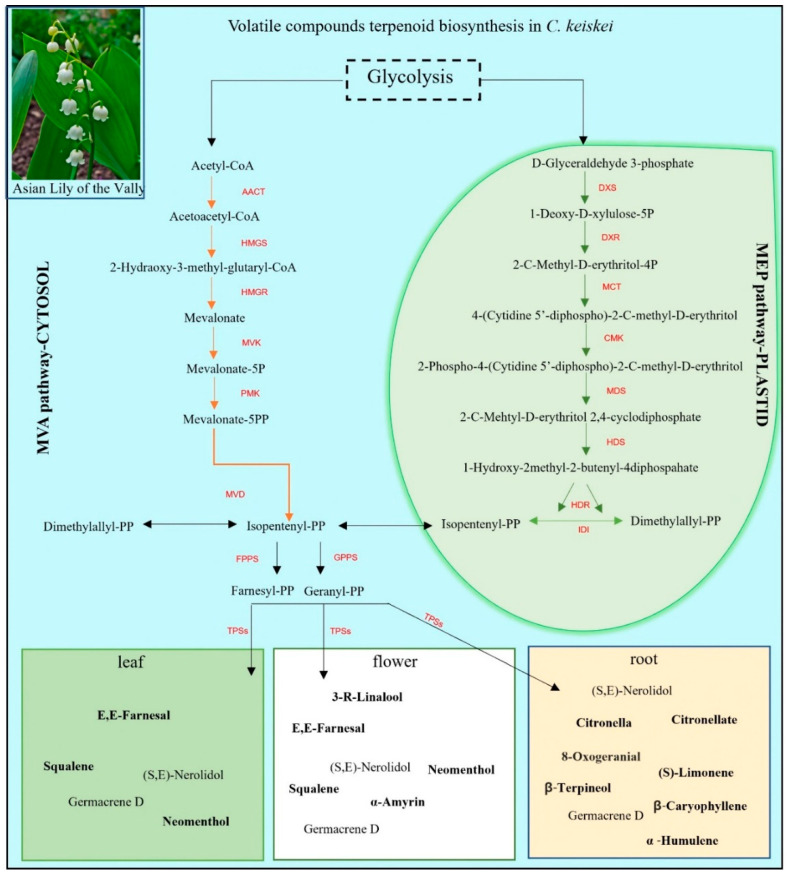
KEGG annotations of the MVA and MEP pathways and their volatile compounds in *C*. *keiskei* (KEGG pathway map reference 00900 and 00909).

**Figure 2 plants-12-02797-f002:**
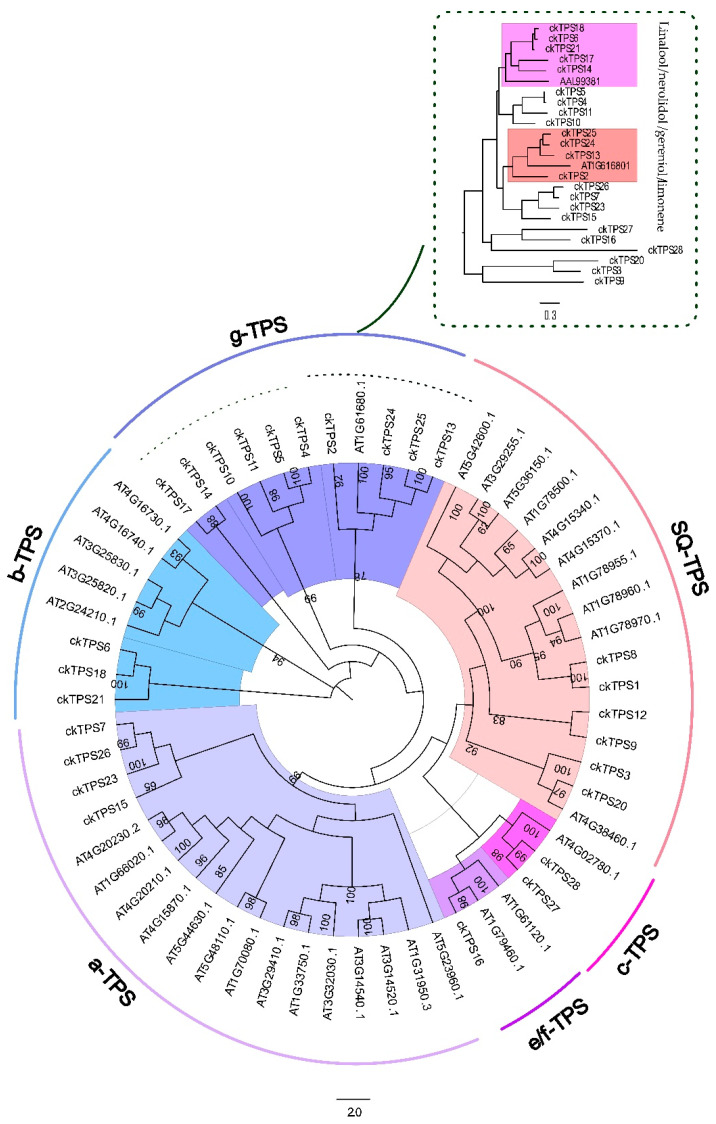
Phylogenetic tree analysis and identification of the *TPS* gene family in *C. keiskei*.

**Figure 3 plants-12-02797-f003:**
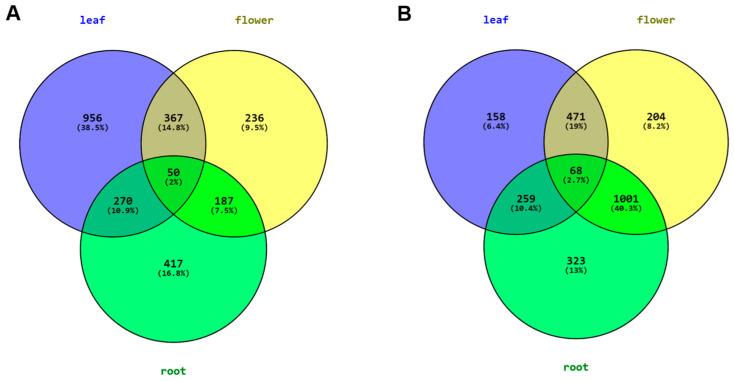
Venn diagram of the overlap of DE unigenes among the leaf, flower, and root of *C*. *keiskei*. (**A**) Upregulated genes and (**B**) downregulated genes.

**Figure 4 plants-12-02797-f004:**
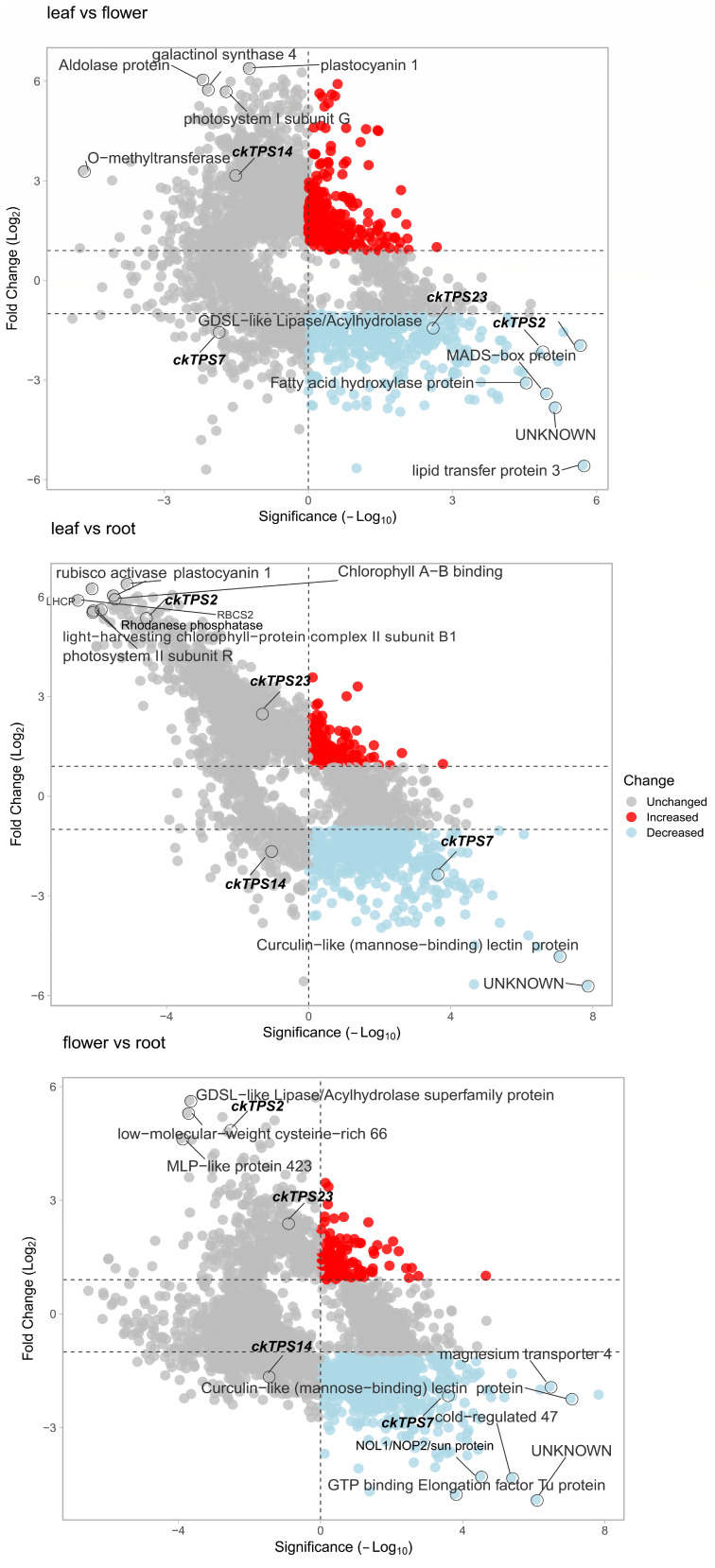
Volcano plot showing comparative differential expression of unigenes and the top four *TPS* genes in *C*. *keiskei*.

**Figure 5 plants-12-02797-f005:**
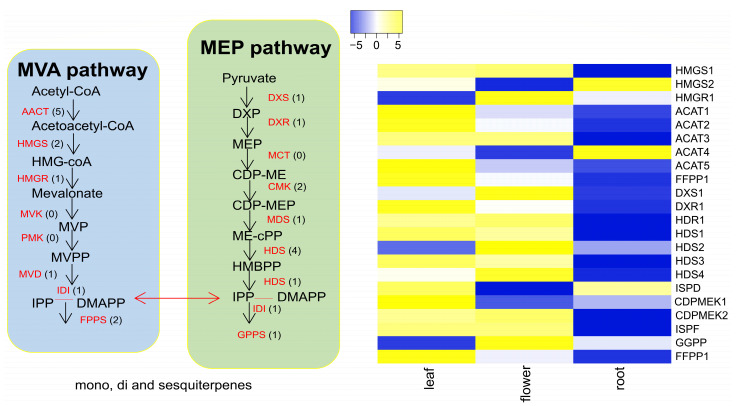
The expression profile of genes in the MVA/MEP (terpenoid backbone) pathway in *C*. *keiskei*. HMGS: Hydroxymethylglutaryl–CoA synthase; *HMGR*: Hydroxymethylglutaryl–CoA reductoisomerase; *ACAT*: Acetyl–CoA C-acetyltransferase; *DXS*: 1–deoxy–D–xylulose–5–phosphate synthase; *DXR*: 1–deoxy–D–xylulose–5–phosphate reductoisomerase; *ISPD* and *ISPF*: 2–C–methyl–D–erythritol 4–phosphate cytidylyltransferase; *CDPMEK*: 4–(Cytidine–5–diphospho)–2–C–methyl–D–erythritol kinase; *HDS*: 4–hydroxy–3-methylbut–2–enyl diphosphate synthase; *HDR*: 4–hydroxy–3–methylbut–2–enyl diphosphate reductase; *GGPP*: Geranylgeranyl pyrophosphate synthase; *FFPP*: farnesyl pyrophosphate.

**Figure 6 plants-12-02797-f006:**
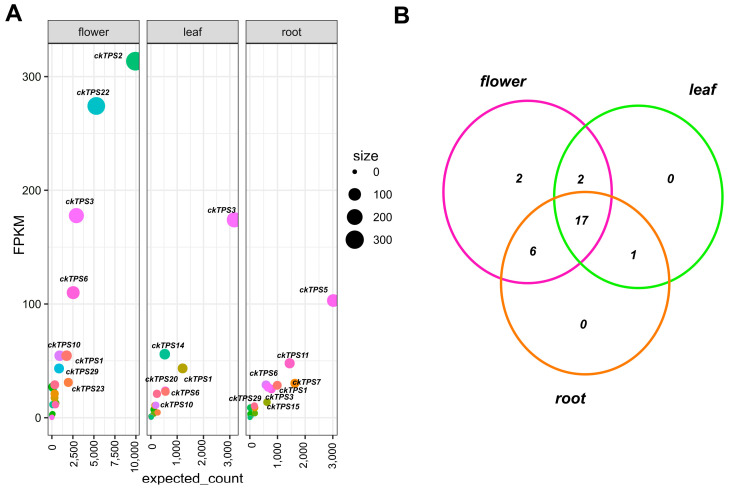
Terpene synthase expression patterns in *C. keiskei*. (**A**). Comparative differential expressed *C*. *keiskei* terpene synthase genes. FPKM: fragment per kilobase of transcript per million fragments mapped, size: FPKM value. (**B**). A Venn diagram of the expected distribution of *ckTPS* genes in *C*. *keiskei* transcriptome.

**Figure 7 plants-12-02797-f007:**
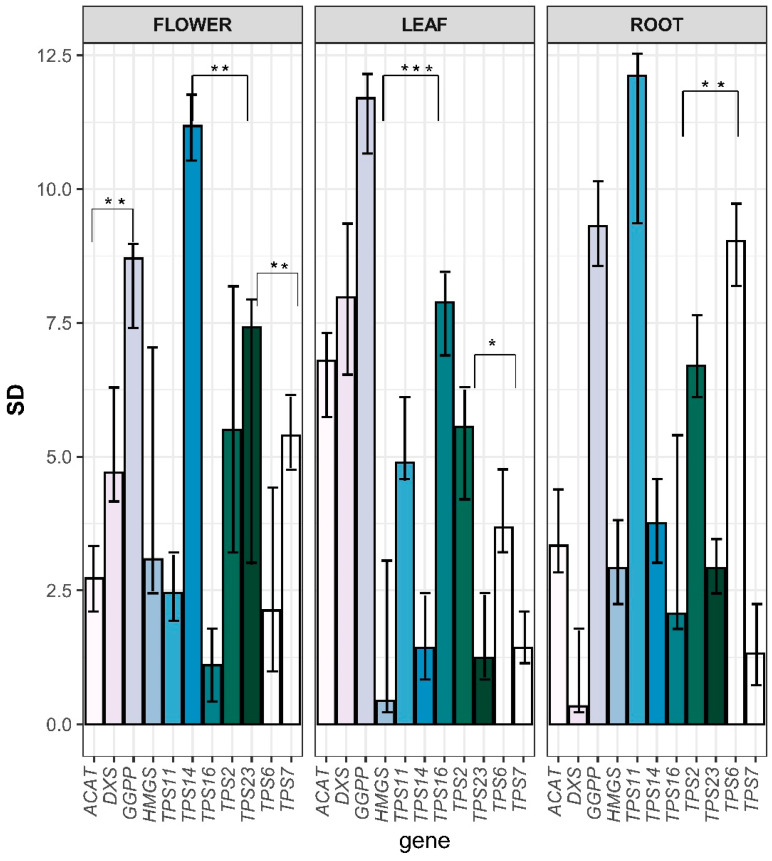
Comparative quantitative analysis of the terpene backbone pathway and *TPS* genes. The error bar indicates the mean of ±SD, and * (*p* ≤ 0.05), ** (*p* ≤ 0.001) and *** (*p* ≤ 0.0001) indicate the significant changes based on *t*-test calibration.

**Table 1 plants-12-02797-t001:** Trinity de novo assembled unigenes and annotation statistics for *C*. *keiskei*.

S. no	*C. keiskei* Tissue	Unigene	N50	TAIR	Pfam	KEGG
1	leaf	72,044	1058.54	14,413	46,578	42,221
2	flower	116,434	972.03	15,679	55,244	50,597
3	root	146,550	802.03	14,880	58,909	45,363
	all	237,129	811.21	33,021	59,121	68,282

## Data Availability

The generated RNA-sequencing database is available from the NCBI-SRA server under accession number SRR20982553-55.

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
