# Peer review of "Comparative Analysis and Identification of Terpene Synthase Genes in Convallaria keiskei Leaf, Flower and Root Using RNA-Sequencing Profiling"

_plants, 2023, doi:10.3390/plants12152797_

Round 1
Author Response
- ‘Introduction’: authors need to represent more statement about mono- and sesqui-compounds in C. keiskei, such as ‘R-linalool, E-nerolidol, germacrene D, trans-ocimene, E-farnesol, neomenthol, nerolidol, squalene, α-terpineol, neomenthol, citronella, limonene, β-terpineol, and β-caryophyllene’, and the difference of content between the leaf, flower, and root of the C. keiskei. As authors presented, TPS gene distribution in the leaf, flower, and root is different, and expression patterns were tissue-specific, so the content of terpenoids might be different between the three tissues.
Response: Thank you. In the introduction section, we have incorporated essential metabolites in the Convallaria keiskei (P. 2, L. 79-86 in the revised manuscript). However, there are no transcriptomic and metabolomic studies to understand the distribution of gene expression and their related metabolites in the different parts of C. keiskei. This is the first study that discussed terpene synthase-related gene expression in the different parts of C. keiskei (P. 14, L. 425-518 in the revised manuscript).
2-in ‘Discussion’: Compounds are the end results of gene expressions. Authors need to reveal the inner relationship between the difference of gene expression in three tissues and the difference of terpenoids content in three tissues, maybe not all of compounds, but as much as possible, which may reinforce the credibility of the results and highlight the significant of conclusion. For example, it is absence of proof about difference of terpenoids compounds and their contents that support Discussion, “page 12-13, line 386-469”.
Response: Thank you for your valuable comments. In the discussion section, we compared and discussed the differentially expressed terpene synthase genes in the C. keiskei and provided the highly expressed genes in Supplementary Table S2 in the revised manuscript (P. 15, L. 513-518).

Reviewer 2 Report
Claude et al. carried out Comparative Analysis and Identification of Terpene Synthase Genes in Convallaria keiskei Leaf, Flower, and Root Using RNA-Sequencing Profiling is an interesting work, Further, the RNA-seq was carried out for three different parts of (C. keiskei), but the transcriptome was simply analyzed, comprehensive analyses are required in this case.
It is also necessary to compare the content of volatile compounds, monoterpenes, and sesquiterpenes in the three parts with the relevant gene expression.
Language has to be improved.
It is also better to show the clear images of experimental materials in Materials and Methods section
Reorganize the Figure 2.
Scientific and gene names should be italicized throughout the text.
Moderate editing of English language required
Author Response
It is also necessary to compare the content of volatile compounds, monoterpenes, and sesquiterpenes in the three parts with the relevant gene expression.
Response: Thank you for your valuable comments. In the discussion section, we compared and discussed the differentially expressed terpene synthase genes in the C. keiskei and provided the highly expressed genes in Supplementary Table S2 in the revised manuscript (P. 15, L. 513-518).
Language has to be improved.
Response: We have submitted our revised manuscript for the English editing service before resubmission.
It is also better to show the clear images of experimental materials in Materials and Methods section
Response: In the revised manuscript, we provided the schematic diagram for the transcriptome assembly method as a supplementary Fig. S1 (P. 3, L. 119-120 in the revised manuscript).
Reorganize the Figure 2.
Response: Fig 2 is an unrooted tree. We could not make a rooted tree to understand the ordering of gene families. Though we have tried the rooted tree it supports the same as an unrooted tree attached below.
The previous TPS phylogeny research also provided similar results as we provided here and does not support subfamily order.
Research articles:
https://onlinelibrary.wiley.com/doi/full/10.1111/pbi.12649
https://www.nature.com/articles/s41598-020-57805-1
https://www.mdpi.com/2073-4425/12/4/518
Scientific and gene names should be italicized throughout the text.
Response: All the gene names are italicized in the revised manuscript.
